# Exploring the Legal Regulation of Social Media in Europe: A Review of Dynamics and Challenges—Current Trends and Future Developments

**Daniele Battista** [1,*] **and Gabriele Uva** [2,*]

1   Department of Political and Social Studies, University of Salerno, 84084 Fisciano, Italy
2   Department of Law, Economics, Management and Quantitative Methods, University of Sannio, 82100 Benevento, Italy
*   Correspondence: dbattista@unisa.it (D.B.); gabriuva@unisannio.it (G.U.)

**Abstract:** Today, the process of digitisation of everyday life pervades all aspects and areas in which human beings move and realise their interests. The political sphere is no exception and is also influenced by technological innovation. Over the last decade, the development of Web 2.0 has meant that cyberspace, albeit through electronic means, has taken on the characteristics of a physical place in the guise of social platforms. Currently, the continued proliferation of social networks is reviving numerous debates and latent issues that are still unresolved. Against this backdrop, research has been undertaken to understand the different aspects and the many meanings of this new dimension across different fields of research. In fact, the work will initially focus on the role they possess in society and the possible negative declinations resulting from disinformation and will then come to a legal overview in terms of European regulations, with reference to the protection of privacy and personal data following the enactment of EU Regulation 679/2016. The objective of this study is to provide a sociological and legal framework for the ethics of artificial intelligence and legal regulation in Europe. This study aims to promote a scientific and political discussion to improve understanding of the pervasiveness of social networks and related legal implications. Additionally, this study seeks to offer a perspective that leads to ethical and sustainable solutions.

**Keywords:** social media; social networks; digital identity; privacy; European regulation

## 1. Introduction

### 1.1. Impact of Technology, Identity, and Social Networks

The extensive technological and conceptual changes affecting our societies raise and pose many questions about how digitisation and the use of social media impact the communities involved. It is widely believed that there is a "crisis" surrounding the use and abuse of contemporary technology and information and communication systems that permeate every aspect of daily life [1]. Considering what is happening in the variegated system in which we are immersed, it is a rather complex exercise to investigate for a full understanding of the phenomenon. The need for this research, without any claim to exhaustiveness, arises by virtue of the state of the art that surrounds us and the now consolidated continuity between the offline and digital dimensions, in which citizens move and act to express their opinions and feelings in a social climate of increasing media personalisation and total multidimensional harmony of life that runs between online and offline [2]. The characteristics of social identity and social networking, before the advent of the net, were limited by the spatial and temporal constraints to which the individual was subject. With the development of the Internet, man has expanded the boundaries of his social networks to the creation of a new social space, cyberspace, which represents the union of the interaction, support, and control of traditional social networks on the one hand, and the characteristics of the Web given by multimedia, content creation, and sharing on the other. They represent, therefore,

the meeting point arising from the use of digital media in various ways—primarily as a tool to support one's own social network but also as a means of expressing one's own social identity and as a means of analysing the social identity of other network members. Having said this, some questions arise spontaneously, the answers to which may help us understand the true relational and communicative scope of social networks on the ongoing digitisation process. In addition, we wonder to what extent and in what way these platforms intervene and work on the construction and definition of an individual's identity, without neglecting what effects they produce in terms of opportunity or risk. Furthermore, we include an examination of the reaction of the European legislature and community jurisprudence on this issue, with an indication of the necessary reform perspectives aimed at protecting the individual and their most intimate sphere.

### 1.2. The Impact of the COVID-19 Pandemic

The penetration of new technologies is perhaps one of the most characteristic and distinctive features of these pandemic years. On several occasions, new terms have suddenly burst into our lexicon and new technological tools have become indispensable. It is a fact that COVID-19 has radically changed methods of communication and daily life, invading fields that were unimaginable until sometime before. This established reality is particularly evident when compared to a few decades ago. The relationship between digitisation and the use of social platforms has reached an impressive combination in recent years, and thus with the advent of social media, a new way of participation has spread within cyberspace [3]. This scenario contributes greatly to putting under the magnifying glass the correlation between the digitisation process and the consequent widening of participation. Indeed, in today's context, digital media takes the form of a new socialisation space [4], through which people learn to disentangle themselves in this new online world [5].

### 1.3. Internet-Centrism and Social Participation

In this new perspective, digital media represents an innovative socialisation channel that helps users to participate in public life, as well as develop new forms of activism. All this could facilitate the involvement of citizens to overcome the existing obstacles in a crisis of collective participation where the transition from an information society to a seduction society is taking place [6]. What is happening at this precise moment in history is symptomatic of a deeper change affecting our society. The pandemic, for instance, has only highlighted some of the critical issues surrounding digitisation, acting as a process accelerator for many other issues. Nevertheless, modern society is strongly influenced by developments related to technological and digital innovation. For decades, the international debate has been concerned with defining and measuring the impact of networks and their consequences in almost all spheres of reality but there is almost always no conclusive data to provide comprehensive answers. So much so that frequently now, when we find ourselves analysing a field as difficult to decipher as this, we refer to a clear semantic field: revolution. The changes taking place during this digital revolution have caused a fundamental paradigm shift; in fact, there are several social dynamics influenced by network technologies. Among these, we cannot fail to consider the process of digitisation that has affected the political world and its landing on social platforms. Generally, the tendency is to attribute to technology an almost mystical ability or an intrinsic purpose for which it is possible, inevitably, to change society and the world again [7]. Sometimes, however, the directions taken seem to be overly simplistic around the debate around digital and innovation. Evgeny Morozov, one of the sharpest observers of networks and contemporary issues has often spoken of 'Internet-centrism'. By this category, Morozov refers to the tendency to consider the net as an actor acting on society from the outside and not as a socio-technical form emerging from within a particular political and social situation. Internet-centrism has the tendency to view technology and the network as a separate sector from the physical world. As if the latter functioned according to its own rules without any social dynamics but only those of a hypothetical and sublime 'cyberspace'. The effort

that needs to be made to address the discussions around technological change, as we said, must go in the same direction, first and foremost eliminating the idea that technologies are always the panacea to the problems that one wants to solve in the 'physical' world. In his essay 'The Internet will not save the world', Morozov also speaks of 'solutionism' as another tendency, specular in many ways to Internet-centrism, which tends to consider it possible to solve complex social or political problems with ready-made and optimisable technological solutions [8].

*1.4. Interconnectivity, Future Perspectives, and the Point of No Return*

A delicate passage that very clearly sets a territory for action in the near future is idealised by Zuboff, who writes: "The digital world is taking over, redefining everything before we are given a chance to reflect and decide. We can appreciate the aids and perspectives that interconnectedness offers us, but at the same time we see new territories of anxiety, danger, and violence opening, as the idea of a predictable future vanishes forever. Today billions of people, of all social strata, ages, and backgrounds, have to answer those questions". This passage underlines, once again, that when it comes to technology, we do not have a predetermined direction towards a desired goal. On the other hand, the combination of behaviours in our lives when we are connected and when we are not connected is now blurred and subtle. Our lives and the hybrid practices between online and offline are called Onlife [9]. The scholar used a metaphor to best describe this term, comparing us to a mangrove society. This is because mangroves live in brackish waters where seas and rivers meet. According to Floridi, in this new existence, there is no longer a difference between online and offline but there is precisely an Onlife, an existence that is as hybrid as the mangrove habitat. We are facing a semantic reversal compared to previous technological evolutions when information influenced technology but not vice versa. Today we are in a context characterised by an increasingly personal environment that creates and manages a multilateral dimension in which we live both online and offline. This case study develops from this context and attempts to trace new ridges of study. In recent years, we have witnessed a proliferation of studies on the possible regulations of social platforms that have become big. The fact remains that the processes of digitalisation that have involved society since the last century, both before and after the advent of the Internet, pose questions that are not purely technological but must be contextualised within the socio-historical processes in which they are produced—hence the decision to open our article with this dissertation that aims to demonstrate how we have now reached a point of no return. In the tradition of studies that, within the sociology of the media, identify the links between the forms of social, technological, and media innovation, it is possible to identify three main paradigms—between them in a relationship of continuity—through which to observe and analyse the forms that society and the relations of individuals with communication take in the digital turn [10]; however, this will not be observable with this research.

## 2. A reconstruction of the Unresolved Situation

As platforms tend to modify and transform interconnectedness in relation to participation, it is worth exploring the dynamics that are disrupting our communities. The new environment is determined by networks well before the development of social platforms [11] and with the spread of new digital technologies, the development of the social environment poses a new ecology of social and political relations between groups and individuals. The ecosystem of communications is strengthened by interactive media as is the level of democracy [12], debate in the extended public sphere is free of intermediaries and involves users that transcend geographical and cultural barriers, thus constructing a new type of citizenship. However, the freedom of the net also entails dangers from its dark side [13]. Not always, however, has there been awareness in society and the scientific community of the damage caused by the "cyber" world. In an initial trend of thinking about this context, as emerged from McGuire's study on the legal or technical perspectives on cybercrime, connectivity to the internet and, more generally, online interaction, it was perceived as an opportunity to improve life and increase rights and freedoms. Only



later did the concept of cyberspace emerge as an unregulated and anarchic space, where connectivity and online interaction constitute one of the greatest dangers to society.

## 2.1. The Impact of Social Media and Technology on the Law

As can be seen from the preceding pages, the impact of social media on society and individuals is of such a magnitude that its influence now spans every sector of public and private life. This is a phenomenon that, evidently, cannot fail to be the subject of legal treatment and attention, study, and analysis by legal practitioners—with particular reference to the impact on subjective legal situations and the relationships that arise from their intersection. It has recently been argued that the advent of technology and its evolution has radically transformed social relations and the everyday life of the individual, to such an extent that it has affected every area of legal science, including that of inheritance. This includes the very ways in which his personality can be realised, not only in the course of his life but also for the time in which he will have ceased to live [14]. The web, through social networks, becomes in fact a social space [15] of relations in which the subject exercises some of his fundamental rights, such as the free expression of thought or free economic initiative; an authentic Societas [16] in which each person builds his own digital identity determined by the peculiar elements of his profile—personal data, political and sexual orientation, networks of friendships, interests, business preferences, images, and information of various kinds.

## 2.2. Digital Identity and Privacy Risks

The use of social media, therefore, realises interests that are worthy of the legal system but, at the same time, exposes the user to risks connected to the protection of his identity and, in particular, of his privacy and personal data. It is well known, on the other hand, that access to the aforementioned digital platforms produces the emergence of questions and problems of a legal nature, in view of the fact that the fundamental resource of social media is linked to the activity of collecting and managing users' personal information [17] and that the main and, tendentially, exhaustive source of regulation of relations on social networks is represented by the agreements stipulated between users and social sites [18]. It seems appropriate, for the purposes of this research and as a methodological premise of the investigation, to dwell on the concept of digital identity. Digital identity means the online representation of a natural person or entity that emerges from a set of data and information containing the characteristics of the interested party and which allow for their identification [19]. Until the second half of the 20th century, identity was only physical, and this identity was taken over by information technology, bending it in an instrumental way for patrimonial use and the pursuit of profit [20]. Digital identity, in a social context where the user subject is induced to share as much information as possible, is composed of a mass of data, such as business or political preferences, and in general by "a world of digital information in which the classifications of data and above all their connection reconstructs an identity that partly matches the real one and partly deforms, magnifies or depresses it" [21]. Digital identity represents, in fact, the key to accessing the digital community and has the function of enabling the identification of the subject within social networks [21]. Entry into social platforms takes place through an initial moment: registration and the construction of one's profile, representative of one's identity. At the same time, access is also conditional on signing the general terms and conditions, which govern the relationship between the parties. At the same time as the signing of the terms of use and service, the privacy policy statement is also submitted for the user's consent, which lists the ways in which data and information will be processed [17]. If we add to this that the regulation of relations with users is placed by social networks through regulations removed from national legal systems with the progressive spread of global non-state law, the picture becomes even more complicated [18]. However, the acquired awareness on the part of legal practitioners and the European legislator has allowed for the affixing of certain limits and the introduction of rules aimed at protecting the weaker party, its identity, its privacy, and the circulation of its data. As noted, the profound evolution of technology that has

marked and modified the intimate essence of personal identity reverberates on the profiles of privacy and confidentiality, which are inseparably connected to it.

## 3. A Still Unclear Perspective

Moreover, the phenomenon of data sharing and collection has increased exponentially and personal data have become a necessary ingredient for the performance of online activities; this is also due to the fact that it is the social users themselves who publish their personal information [19]. The most obvious consequence, and one that raises questions about the needs and methods of protection for jurists, is the increased risk that the invasive potential of the telematic information system in human existence will become a tool for sophisticated forms of social control [22]. In addition, two-and-a-half decades after the birth of the Internet, it is clear that the landscape of cyber security threats has changed dramatically. The maturity of social media has made it more difficult to fight cybercrime, as network technologies have brought about a fundamental transformation of criminal behaviour and a very different organizational logic from that of offline organized crime [23].

### 3.1. The Evolution of the Concept of Privacy and European Legislation

From this perspective, it is interesting to analyse the evolution of the concept of privacy: from the protection of the personal and intimate sphere of the individual to the right to maintain control over one's own information and data to dominate their circulation [24]. The investigation aims, on the one hand, to understand whether the relationship between the right of access to the network—to social media—and the right to privacy with regard to the circulation of data on Web 2.0 is sustainable, and, on the other hand, whether the legislative interventions enacted so far have been able to strike a balance and attempt to indicate a possible reform perspective that is more functional to the multiple, as we have seen, interests in the field. With the approval of Directive 95/46/CE, the subject of which is the protection of individuals regarding the processing of personal data, the protection of personal data entered fully into European law. In Recital No. 2 of that directive, it was stated that "data processing systems are at the service of mankind; that they must, whatever the nationality or residence of natural persons, respect their fundamental rights and freedoms, notably their privacy, and contribute to economic and social progress, the development of trade and the well-being of individuals" [25]. Already at the time, there was a clear desire to imagine a regulation that was functionally connected and oriented towards sustainability between the requirements of the free market and free competition and those of human dignity and the development of human personality. In 2000, not by chance, with the approval of the Charter of Fundamental Rights of the European Union, it was affirmed in Article 8 that everyone has the right to the protection of personal data concerning him or her, as well as the right to access and rectify the data collected [26]. Later, with the entry into force of the Lisbon Treaty, the right to the protection of personal data enjoyed by every individual was reaffirmed in the TFEU. The evolution of the historical-legislative level of privacy legislation can be seen in Table 1 below.

**Table 1.** Historical-legislative evolution of privacy legislation.

| Year | Event |
|------|-------|
| 1950 | Article 8 of the European Convention on Human Rights (ECHR), which establishes the right to respect for private life |
| 1981 | The Council of Europe adopts Convention 108—now Convention 108 Plus—which is the largest European-level document for the protection of personal data |
| 1995 | The EU adopts Directive 46/95 on the protection of personal data |
| 2001 | Approval of the Nice Charter, in which Article 8 establishes the right to protection of personal data |
| 2007 | The right to the protection of personal data enjoyed by every person was reaffirmed in the TFEU. In addition, the legislative competence of the European Parliament and the Council on the subject was established |
| 2016 | General Data Protection Regulation (GDPR) |

### 3.2. Reasons for Reform

In light of what emerges, it becomes clear that regulating social platforms is crucial to ensure that they are used fairly and that citizens' fundamental rights, such as privacy and freedom of expression, are protected [27], also in view of the reality in which we find ourselves, which sees social platforms invaded by a wide range of threats such as the spread of false information combined with a massive dose of hate and discrimination content. The legislative competence of the European Parliament and the Council on the subject was also established [28]. Nevertheless, the rapid evolution of technology and the established pervasiveness of social networks led the European legislator to an intervention aimed at regulating and containing the significant extent of personal data collection and sharing. The need for reform has arisen because "Directive 95/46/EC has not prevented the fragmentation of the application of personal data protection across the territory of the Union, nor has it eliminated legal uncertainty or the widely held public perception that online operations in particular pose risks to the protection of natural persons" [29] and because of the continuous evolution of the concepts of privacy and data protection resulting from the progress of online services [23].

### 3.3. European Regulation No. 679/2016: Merits and Limitations

The reform occurred with the issuance, on 27 April 2016, of European Regulation No. 679/2016 on the processing of personal data, which repealed and replaced the previous directive. The provision is applicable to any form of processing of personal data, except for the peremptory cases concerning reasons of national or common security, as well as for purposes of prevention, investigation, detection, or prosecution of criminal offences. Moreover, in the typical drafting style of the acts of the European legislator, it is stated that personal data means any information concerning an identified or identifiable natural person (the data subject). Instead, processing is defined as "any operation or set of operations which is performed upon personal data or sets of personal data, whether or not by automated means, such as collection, recording, organisation, structuring, storage, adaptation or alteration, retrieval, consultation, use, disclosure by transmission, dissemination or otherwise making available, alignment or combination, restriction, erasure or destruction." However, apart from the merely definitional aspect, it seems appropriate for the purposes of this research to go into the substance of the regulation in order to grasp the interests that the legislation tends to protect and the balance that is achieved or configured with the protection of the individual and his personal sphere. Article 5, in particular, states that processing must be carried out lawfully, fairly, and transparently, as well as for specified, explicit, and legitimate purposes and in a manner compatible with those purposes. Lawfulness, as careful doctrine has pointed out, does not end within the regulation but implies compliance with the rules of the system as a whole and not only with data protection laws [30]. The principle of lawfulness is closely connected with the consent of the data subject and the necessity of the processing, as is clear from the literal scope of Article 6. Fairness, on the one hand, concerns both good faith in relation to the data subject and compliance with ethical and deontological standards. Transparency, on the other hand, ensures the subject's awareness, since only if he or she receives crystal-clear information is he or she able to take a decision and give informed consent. Consent plays a crucial role in the legislation protecting privacy and personal data. Consent is deemed to be any manifestation of the data subject's free, informed, specific, and unambiguous willingness to consent to the processing of personal data.

### 3.4. The Importance of Consent

On the other hand, an analysis of the legislation and the way in which social networks are accessed highlights structural and functional shortcomings of consent. First of all, it is useful to note how consent is only defined but not also fully regulated by the regulation. Moreover, the regulation refers to 'any' manifestation of will, which leads to a wide interpretation of its application. However, the greatest perplexities of a technical-legal

nature arise when reading Article 7(4) of the aforementioned regulatory act: "in assessing whether consent has been freely given, the utmost consideration shall be given to the possibility, inter alia, that the performance of a contract, including the provision of a service, may be made conditional on the provision of consent to the processing of personal data not necessary for the performance of that contract". The provision refers to the case, very frequent for certain online services and social networks, in which the user is faced with an ultimatum: either he gives consent or he cannot use the service. This consent also covers, inter alia, the processing of data not necessary for the provision of the service but also the transmission of those data to third parties for commercial purposes. In these cases it is presumed, both by virtue of Article 7(4) of the regulation and recital 43 thereof, that consent has not been freely given. On this point, it has been stated, in fact, that "if consent is a non-negotiable part of the general terms and conditions of the contract/service, it is presumed not to have been freely given" [23]. The approach of the European legislator is to consider it reasonably plausible that, in the above-mentioned cases, consent is not freely given but conditional on the possibility or otherwise of using the service. The issue is made even more burdensome and complex, in terms of lawfulness and merits, when, in order to access the social platform, one accepts with a single click both the general terms and conditions and the processing of personal data. In fact, consent, to be specific, as stated in the definitions in Article 4, should be given for each individual processing. This is lacking in social networks because although multiple purposes are referred to in the privacy policies, different specific consents do not correspond to them [31]. Furthermore, free and unambiguous consent requires adequate and accessible information. It has been argued that the information preparatory to consent assumes the role not so much of bridging information asymmetries in the relationship between two subjects but above all enabling control over the processing process and the rights due to the data subject during the processing itself [32].

*3.5. Big Data and Data Monetization*

However, in addition to the perplexities of a technical-legal nature, there is a further element of reflection: today's society is increasingly dependent on the analysis of an enormous amount of data and the data has become a currency and has taken on a patrimonial value [33]. There are those who theorise, in this regard, that the processing of personal data has shifted from a moral dimension of protection of a fundamental right to a negotiation dimension of data commercialisation. The former sees in the data an explication of identity and the consequent correct processing of personal data, whereas the latter considers the data susceptible to exchange having patrimonial value [34]. In fact, as careful doctrine has noted [35], it has long been the case that social operators offer users services that are apparently free of charge but which in reality are financed by users' personal data and their re-use for advertising purposes. This approach is also grounded in some regulatory data and in a ruling of the Italian Supreme Court. First of all, Recital No. 47 of EU Reg. 679/2016 states that processing of the data subject for direct marketing purposes may constitute a legitimate interest. Add to this the EU Directive 2019/770 on contracts for the supply of digital content and digital services, which expressly states the consideration nature of personal data in relation to the use of digital services [36]. In 2018, the Italian Court of Cassation intervened, for the first time, about the exchange of personal data, stating that "the legal system does not prohibit the exchange of personal data, but nevertheless requires that such an exchange be the result of a full and in no way coerced consent" [37]. This pronouncement recognises, therefore, that personal data can be exchanged and intervenes in a debate, never dormant and still ongoing, on which this contribution attempts to shed light. It seems appropriate to recall other pronouncements, in order to underline the importance of the issue and the now widespread awareness, among legal practitioners, of the need to place a limit on the so-called negotiability of personal data. In particular, the United Sections of the Italian Court of Cassation, with a precise reconstruction of the national and supranational jurisprudential elaboration, reaffirmed that the right to information does not automatically equate to the right to new and repeated disclosure of personal data [38].

The profound link between the protection of personal data and the rights enshrined in the ECHR is a decisive factor in the pronouncements of the Court of Justice of the EU which, over the years, has interpreted the right to the protection of one's own data extensively, reconstructing it in the light of the fundamental rights of the individual [39]. This latter approach appears to be shareable but the element that sets a limit to the intervention of the legislator and the case law is represented by the fact that the citizen-user, although aware of the risk that his data are not processed in a correct and lawful manner, would seem not to care in order to access the service and enjoy its benefits and services.

*3.6. Artificial Intelligence and the Challenges of European Regulation*

In the face of a rapidly changing world, driven by unprecedented technological and digital transformation, the regulation of social platforms has become essential to ensure their safe and responsible use for users [40]. However, their regulation is a complex and constantly evolving topic, as it must take into account ongoing processes. Despite its essential nature, regulation must constantly be adapted to accommodate new challenges and opportunities presented by technology. For example, with the increased use of artificial intelligence and blockchain technology, the legal regulation of social platforms must be calibrated to best address these unprecedented nuances. It is well known that artificial intelligence is changing the nature of human experience and our moral lives and we must be aware of its effects on our individual and collective ethics [41]. Defining artificial intelligence is a difficult task, and indeed many definitions have arisen over the years. In the opinion of the European Commission, the expression refers to systems that display intelligent behaviour by analyzing their environment and taking actions, to a certain degree of autonomy, to achieve specific goals [42]. The European Union, facing this new epochal challenge, has decided to intervene with a proposal for AI regulation, currently under discussion, through a horizontal approach in order to protect the digital sovereignty of the Union and make use of its regulatory tools and powers to shape global rules and norms and become a global reference on the subject by becoming a leader in norm-production [43]. At the opening of the proposal, it is stated that the Commission also wants to achieve the following specific goals through the regulatory tool. Artificial intelligence is a technology that is changing the world in a significant and unprecedented way. It is important that the European Union takes a proactive approach to address the challenges and opportunities offered by this technology. Firstly, it is crucial to ensure that the artificial intelligence systems placed on the market are safe and comply with existing legislation regarding fundamental rights and the values of the Union. Additionally, it is important to ensure legal certainty to encourage investments and innovation in AI. To do this, it is necessary to improve the governance and effective implementation of existing legislation regarding fundamental rights and security. Finally, the European Union must facilitate the development of a single market for lawful, safe, and reliable AI applications and prevent market fragmentation. Aside from legitimate aspirations for normative leadership and goals to be achieved, one of the reasons that prompted the European Union to intervene is the consideration that dangers are to be prevented in the opacity, unpredictability, and a certain degree of autonomy of some AI systems [44]. However, as recent doctrine has observed, the intrinsic limit connected to the horizontal approach is that "since the regulations are not aimed at resolving specific problems or filling in certain legal gaps, they must necessarily be applicable to any sector, both in the health sector and in the financial sector. Therefore, not ad hoc rules to solve a particular problem or remove legal obstacles, but general provisions to outline a comprehensive framework, a reference context in which AI systems will operate, even those yet to come. Pending a deeper and more analytical analysis of the European legislation on AI, it appears appropriate to share an aspiration from attentive doctrine that is the need of facing "quantum historical leaps," to restore a social balance through a wise exercise of power and effective control, aimed at seeking the general interest and protecting human dignity [45].

### 4. Conclusions

In conclusion, it can be said that a series of questions emerge from this work which, in part, have been answered in some regulatory or jurisprudential interventions and which still pose problems of a legal nature. The search for a sustainable balance point, that is, the point at which the reasons of economic operators intersect and meet with the reasons for protecting the individual and his fundamental rights, which are closely connected with his dignity, necessarily passes through a growing awareness on the part of users of the importance of their personal data and a broader, more widespread and decisive legislative intervention. In conclusion, effective regulation of social platforms should balance the protection of fundamental rights of users, such as privacy and freedom of expression, with the needs of the platforms themselves to provide services to their users and earn profits. The challenge lies in finding a balance between these competing interests.

*Perspectives of Reform*

In this sense, we believe that a European reform on the issue of personal data protection in the use of social platforms is necessary. In our opinion, data must assume a purely moral function and not even a negotiating one, unless it is necessary and indispensable to use that particular social or app, by means of a new regulation that avoids speculative attitudes of social networks that prevent access to their functions if the subject does not accept certain contractual conditions regarding the information to which these platforms will have access. Furthermore, it seems to us necessary that consent should be truly informed and that the language of the disclosures should be clear and comprehensible to anyone, as well as succinctly set out, even with the aid of graphics and audio-visual content, and that the disclosures should not be viewable by referring to links or other navigation windows. Furthermore, consent must not be considered as any manifestation of will but must be outlined and defined with a clear notion, strict in language and free of general clauses so that it is not subject to interpretation. In addition to the solutions mentioned above, there are other considerations that could be taken into account in the context of social platform regulation. Firstly, education with regard to certain issues. Users could be educated on issues related to privacy and online security so that they are better able to protect their personal data and make informed choices about the use of social platforms. Decentralization could also be useful, meaning that user groups are managed instead of individual companies, in order to reduce dependence on algorithms and ensure greater transparency in the management of user data. Obviously, all this is also linked to cooperation between public authorities and social platforms, which could collaborate to define a regulatory framework that protects user privacy and promotes online security. This scenario of assumptions identified after our analysis fits into a prospective international regulatory framework since these platforms have a global reach and their regulation varies from country to country. The regulation of social platforms requires, in summary, a deeper reflection on many complex issues and a balance between the needs of users, platforms, and public authorities.

**Author Contributions:** Formal analysis, G.U.; Investigation, D.B. All authors have read and agreed to the published version of the manuscript.

**Funding:** This research received no external funding.

**Institutional Review Board Statement:** Not applicable.

**Informed Consent Statement:** Not applicable.

**Data Availability Statement:** Not applicable.

**Conflicts of Interest:** The authors declare no conflict of interest.

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
