# Peer review of "Exploring the Legal Regulation of Social Media in Europe: A Review of Dynamics and Challenges—Current Trends and Future Developments"

_sustainability, doi:10.3390/su15054144_

Round 1
Reviewer 1 Report
Review: Social media and legal regulation on the European continent: an evolving phenomenon. Some comments are given as follows:
1. The introduction should be divided into several parts. The main contributions should be further highlighted.
2. Also, the Section 3 should be divided into several parts. It is so clear the main results of Section 3.
3. This work lacks figures and tables. If possible, please add related figures or tables.
Author Response
Dear Reviewer, please see the author's response on the attachment.

Reviewer 2 Report
At the outset, I applaud the authors for articulating these concerns. In my opinion, this article is highly relevant and timely. The articulation/flow is good. However, at the same time, I am a bit confused. For instance, this article is not a standard research article - this is more like a white paper or editorial. Who is target audience of this article? Scholars or Policymakers or Common users - not clear! Also, the length of the paper is significantly shorter than a typical research paper. So, I am not sure whether this article should be accepted or not. Probably, the editor can take the call regarding the same.
I have two suggestions:
1. This paper's takeaway is unclear in the initial few pages. So, if the authors can articulate the same in the beginning (i.e., introduction section), it will be easier for readers.
2. A plethora of papers pointed out these issues. For instance, "the ethics of AI" is a vibrant research domain. Thus, the anchoring of the paper with the academic literature, as well as policy-level measures, can be further improved.
Author Response

(The authors gave the same response as above.)

Reviewer 3 Report
Thank you for submitting your work for evaluation.
Your work discussed how the public, political, and industrial spheres are influenced by technological innovation. Over the last decade, the development of Web 2.0 has meant that cyberspace, albeit through electronic means, has taken on the characteristics of a physical place in the guise of social platforms, you suggested. Against this backdrop, your research attempted to understand the different aspects and meanings of new dimensions resulting from blending cyberspace and the physical sphere across various research fields.
However, it has many flaws. I offer my advice listed below to improve your work.
1. NO PARAGRAPHING THROUGHOUT THE PAPER
The entire introduction is a single paragraph from pages 1-3. Break it into about 7-8 paragraphs to help your readers follow the logic of your argument.
There are four essential elements that an effective paragraph should consistently contain: unity, coherence, a topic sentence, and sufficient development. A good paragraph should do the following:
· presents a single idea
- begins with a topic sentence that makes this single idea evident
- contains support in the form of sentences that convey this single idea
- is strategically organized to maintain the flow
- maintains your paper’s objective
- informs and entertains your reader about your paper’s overall idea.
2. “2. A reconstruction of the unresolved situation” section is also a single paragraph. Please divide this section into about four paragraphs.
3. Section “3. A still unclear perspective,” from pages 4-7, is also a paragraph. Divide this section into eight paragraphs.
4. “4. Conclusion” section should be about two paragraphs, not one.
The paper is difficult to read due to a lack of paragraphing. Still, I have been constructive, thorough, and kind.
5. Apart from a lack of paragraphing that made it difficult to follow the logic of your argument, your paper will benefit from integrating conceptualizations of how the public, political, and industrial spheres are influenced by technological innovation and the development of Web 2.0. Against this backdrop, your research attempted to understand the different aspects and the many meanings of new dimensions resulting from the blending of cyberspace and the physical sphere across various fields of study. Therefore, I encourage you to engage with these conceptualizations and integrate them into discussions of your work to boost your contribution.
For example, see the semantic (legal) or syntactic (technical) perspectives on cyber criminality, as theorized by McGuire (2017).
See the “Geopolitical, Socioeconomic and Psychosocial classifications” of cyberspace proposed by Ibrahim (2016).
See multiple categorizations of cybercrime and cyberspace proposed by Wall (2013).
See the Tripartite Cybercrime Framework (TCF), further developed by Lazarus, Button, and Kapend (2022).
Reference:
McGuire, M. (2017). Cons, constructions, and misconceptions of computerrelated crime:
From a digital syntax to a social semantics. Journal ofQualitative Criminal Justice and
Criminology , 6(2), 137–156. https://doi. org/10.21428/88de04a1.505d151e
Lazarus, S., Button, M., & Kapend, R. (2022). Exploring the value of feminist theory in
understanding digital crimes: Gender and cybercrime types. The Howard Journal of Crime
and Justice, 61(3), 381-398. https://doi.org/10.1111/hojo.12485
Wall, D. (2007). Cybercrime: The transformation of crime in the information age (Vol. 4).
Polity

Author Response

(The authors gave the same response as above.)

Round 2
Reviewer 1 Report
The authors have carefully revised. I have no further comments.
Reviewer 2 Report
Thanks to the authors for your efforts in revising the manuscript significantly. My earlier comments have been mostly addressed and I have no problem recommending this article for publication.
Reviewer 3 Report
(A)
Following my advice on paragraphing, the structure of your paper has significantly improved!
(b)
Your work has shown improvement. However, I suggest you comprehensively integrate all three recommended works on cybercrime and cyberspace categorizations to enhance it further.
Why does it matter one might ask? Each of these works offers unique insights and perspectives that could have enriched your analysis. Each work offers a unique perspective, and combining them would result in a more comprehensive and nuanced analysis.